# Physical Activity and the Risk of COVID-19 Infection and Mortality: A Nationwide Population-Based Case-Control Study

**DOI:** 10.3390/jcm10071539

**Published:** 2021-04-06

**Authors:** Dong-Hyuk Cho, Sun Ju Lee, Sae Young Jae, Woo Joo Kim, Seong Jun Ha, Jun Gyo Gwon, Jimi Choi, Dong Wook Kim, Jang Young Kim

**Affiliations:** 1Division of Cardiology, Department of Internal Medicine, Yonsei University Wonju College of Medicine, Wonju 26426, Korea; why012@yonsei.ac.kr; 2Department of Big Data Strategy, National Health Insurance Service, Wonju 26464, Korea; ju6801@nhis.or.kr (S.J.L.); haagoon@nhis.or.kr (S.J.H.); 3Department of Sport Science, University of Seoul, Seoul 02504, Korea; syjae@uos.ac.kr; 4Division of Infectious Disease, Department of Internal Medicine, Korea University College of Medicine, Seoul 02841, Korea; wjkim@korea.ac.kr; 5Division of Transplantation and Vascular Surgery, Department of Surgery, Korea University College of Medicine, Seoul 02841, Korea; doctorgjg@gmail.com; 6Division of Endocrinology and Metabolism, Department of Internal Medicine, Korea University College of Medicine, Seoul 02841, Korea; jjimchoi@gmail.com

**Keywords:** severe acute respiratory coronavirus 2, COVID-19, physical activity, mortality

## Abstract

Regular physical activity (PA) is known to reduce the risk of serious community-acquired infections. We examined the association of PA with the morbidity and mortality resulting from coronavirus disease (COVID-19) infection in the South Korean population. Patients who tested positive for severe acute respiratory coronavirus 2 and who underwent public health screening between 2014 and 2017 (*n* = 6288) were included. Age- and sex-matched controls (*n* = 125,772) were randomly selected from the Korean National Health Insurance Service database. Leisure-time PA was assessed using a self-reported questionnaire. The mean PA levels were lower in the patient than in the control group (558.2 ± 516.3 vs. 580.2 ± 525.7 metabolic equivalent of task (MET)-min/week, *p* = 0.001). Patients with moderate to vigorous PA (MVPA) were associated with a lower risk of COVID-19 morbidity (odds ratio (OR), 0.90; 95% confidence interval (CI), 0.86–0.95). In addition, a standard deviation (SD) increment in MET/week (525.3 MET-min/week) was associated with a 4% decrease in the risk of COVID-19 morbidity (OR, 0.96; 95% CI, 0.93–0.99). MVPA and an SD increment in MET/week were associated with lower mortality (MVPA: OR, 0.47; 95% CI, 0.26–0.87; per SD increment: OR, 0.65; 95% CI, 0.48–0.88). Higher levels of regular PA were associated with a lower risk of COVID-19 infection and mortality, highlighting the importance of maintaining appropriate levels of PA along with social distancing amid the COVID-19 pandemic.

## 1. Introduction

Coronavirus disease 2019 (COVID-19) is an infectious disease caused by severe acute respiratory syndrome coronavirus 2 (SARS-CoV-2) [1]. Mounting COVID-19-positive cases and deaths are a public health challenge; the World Health Organization (WHO) declared COVID-19 a global pandemic in March 2020 [2]. Currently, there is no cure for COVID-19, except for supportive treatments. Quarantine for a defined period is currently the best option to reduce the spread of COVID-19 [3]. Thus, national and regional governments have implemented social distancing; restricted movements; and mandated business closure, quarantine, and isolation. Amid this pandemic, quarantine has resulted in a change in human behaviour, including physical activity [4]. A worldwide study with 455,404 subjects, using smart-phone accelerometers, revealed that the mean daily step count had significantly decreased by 27.3% within 30 days after the declaration of the pandemic by the WHO [5].

In fact, the importance of physical activity in decreasing the risk of adverse health conditions, including non-communicable diseases, has been established [6]. Furthermore, physical activity has been shown to reduce the risk of serious community-acquired infections as well as mortality [7]. The Wisconsin Upper Respiratory Symptom Survey revealed that frequent physical activity reduced the risk of upper respiratory infections by 43% [7]. Additionally, regular exercise decreases the risk of influenza-associated mortality [8]. More recently, emerging evidence has highlighted the association between lifestyle risk factors and COVID-19 [9]; for example, individuals with lower levels of physical activity or exercise capacity were at an increased risk for COVID-19-related hospitalisation [9,10]. Furthermore, in a study that included 215 patients positive for COVID-19, physically active patients demonstrated a faster recovery time than non-physically active patients [11].

However, in the context of the on-going pandemic, one key question that remains unanswered is whether regular physical activity has a protective effect against SARS-CoV-2 infection and COVID-19-related mortality. Based on the available information, it is hypothesised that high levels of physical activity can reduce the risk of infectious diseases. To quantify the evidence, we examined the associations between regular physical activity and the risk of COVID-19 infection and mortality in a nationwide case-control study of the South Korean population.

## 2. Materials and Methods

### 2.1. Data Sources and Target Population

This was a retrospective, nationwide study that included all Koreans who tested positive for SARS-CoV-2, aimed at evaluating the characteristics and outcomes of patients with COVID-19. The Republic of Korea was one of the first countries to experience a nationwide outbreak of COVID-19 [12]. The Korean Ministry of Health has supplied complimentary and mandatory public health insurance coverage for all patients with COVID-19, amid the pandemic. The Korea Disease Control and Prevention Agency assembled the study population based on an epidemiological investigation. The Korean National Health Insurance Service (NHIS), Korea Disease Control and Prevention Agency, and the Health Insurance Review and Assessment Service facilitated this study and provided the electronic medical records of the patients. Thus, demographic information, previous health screening results, and all healthcare records within the last 5 years were accessible, along with test results for SARS-CoV-2 and clinical outcomes of COVID-19. A detailed study protocol has been previously described [13].

Based on reports of exposure to COVID-19, and physical examinations, the attending physicians performed the SARS-CoV-2 tests. Among a total of 13,612 consecutive patients with COVID-19 between 1 January and 16 July 2020, we selected 6286 patients who underwent NHIS public health screening between 2014 and 2017. This study implemented the ethical standards of the 1964 Declaration of Helsinki and its later amendments. Written informed consent for the present study was waived as this study relied on retrospective analysis.

### 2.2. Case Patients and Controls

The SARS-CoV-2 infection was confirmed using the real-time reverse transcriptase polymerase chain reaction of pharyngeal and nasal swab samples, in accordance with the WHO recommendations [14]. Case patients were defined as the individuals over 18 years old with positive results for the SARS-CoV-2 test. For every case patient, we selected up to 20 random controls from the target population who underwent the national health screening. COVID-19 related mortality was defined as the termination of isolation due to death. Figure 1 represents the flow chart of the study.

### 2.3. Physical Activity and Other Variables

The Korean NHIS recommends health screening biennially for all Koreans over 40 years of age or annually for manual workers over 18 years of age. This screening process allows for the medical history, anthropometric parameters, and biochemical laboratory tests of individuals. Furthermore, we investigated specific risk factors, such as physical activity, alcohol consumption, and history of smoking using a standardised questionnaire [15].

The questionnaire for physical activity comprised three parameters to determine frequency (days in recent 7 days, at least 30 min per day): (1) light intensity: walking at own pace at slow speed; (2) moderate intensity: brisk walking, playing tennis, or slow cycling; and (3) vigorous intensity: running, jogging, climbing, or bicycling or fast cycling. Physical activities of light-, moderate-, and vigorous- intensities were rated as 2.9, 4.0, and 7.0 metabolic equivalent of tasks (METs), respectively. We calculated the energy expenditure of the subjects by adding the frequency and intensity of physical activity, as previously described [16,17]. The level of physical activity was grouped according to the energy expenditure into totally sedentary or physical inactivity = 0, physical activity < 500, 500 ≤ physical activity < 1000, 1000 ≤ physical activity < 1500, and physical activity ≥ 1500 METs-min/week.

### 2.4. Statistical Analyses

We performed propensity score matching between the case and control groups using logistic regression after adjusting for age and sex. Continuous variables are presented as means ± standard deviations (SDs) and categorical variables as frequencies and percentages. In addition, we compared the differences in demographic, clinical, and laboratory variables between the case and control groups using the chi-square test or Student’s *t*-test. Multivariable logistic regression was used to assess the independent association between physical activity and COVID-19 after adjustment for continuous variables (age, alcohol consumption, and economic income), dichotomous variables (sex, obesity, hypertension, diabetes mellitus, dyslipidaemia, ischaemic heart disease, and stroke), and smoking (current/former/never). We constructed two adjusted models. In model 1, we adjusted age and sex. Model 2 included economic income, medical history (obesity, hypertension, diabetes mellitus, dyslipidaemia, ischaemic heart disease, and stroke), smoking status, and alcohol consumption in addition to the variables used in model 1. Results are expressed as odds ratios (ORs) with 95% confidence intervals (CIs). Statistical analyses were performed using the SAS version 9.4 software (SAS Institute Inc., Cary, NC, USA). A two-sided *p* value < 0.05 was considered statistically significant.

## 3. Results

### 3.1. Baseline Characteristics

There were 6288 case patients with positive results for SARS-CoV-2 and 125,780 matched controls. Table 1 shows the comparisons of the baseline characteristics of the case patients and controls. These comparisons revealed no difference in the mean age, sex, and waist circumference of the subjects. Body mass indexes were higher in case patients than in the controls. The case patients had a higher prevalence of obesity and diabetes mellitus with a lower prevalence of hypertension.

### 3.2. Physical Activity and the Risk of COVID-19

The mean level of physical activity was 579.1 ± 525.3 MET-min/week for all subjects. The proportion of moderate to vigorous physical activity (MVPA) was lower in the case patients than in the controls (*p* < 0.001) (Table 2). The mean physical activity level was lower in the case patients than in the controls (558.2 ± 516.3 MET-min/week vs. 580.2 ± 525.7 MET-min/week, *p* = 0.001).

MVPA was associated with a lower morbidity in COVID-19 patients (adjusted OR, 0.90; 95% CI, 0.86–0.95). Figure 2 and Table 2 show the linear association between physical activity and the morbidity of COVID-19 patients. According to the level of physical activity in MET-min/week, the morbidity of COVID-19 patients linearly decreased (*p* for trend = 0.002). Every 1 SD of MET-min/week was associated with a 4% lower morbidity in COVID-19 patients (adjusted OR, 0.96; 95% CI, 0.93–0.99).

### 3.3. Physical Activity and the Risk of Mortality

A total of 92 deaths occurred during the isolation period due to COVID-19. MVPA was associated with a lower mortality in COVID-19 patients compared with the physically inactive group (adjusted OR, 0.47; 95% CI, 0.26–0.87) (Table 2). The mortality of COVID-19 patients linearly decreased (*p* for trend = 0.007) with an increase in the level of physical activity. In the linear analysis, the level of physical activity in SD increments in METs-min/week was associated with a lower mortality of COVID-19 patients (adjusted OR, 0.65; 95% CI, 0.48–0.88).

### 3.4. Subgroup Analysis

Figure 3 illustrates the risk of COVID-19 associated with physical activity, according to the presence of various variables. Central obesity significantly affected the influence of physical activity on the risk of COVID-19. The beneficial effects of physical activity were more significant in subjects with central obesity than in non-obese subjects (*p* for interaction <0.001). However, age, general obesity, hypertension, and diabetes mellitus did not modify the influence of physical fitness on the risk of COVID-19.

## 4. Discussion

This study revealed that MVPA was associated with a 10% lower risk of COVID-19 infection and a 53% lower risk of COVID-19 infection-related mortality, independent of confounding factors. Furthermore, the highest quintile of physical activity (≥1500 MET-min/week) was associated with a 25% and 77% lower risk of COVID-19 infection and morality, respectively, compared with the physically inactive group. Similarly, an SD increment in MET-min/week was associated with a 4% and 35% lower risk of COVID-19 infection and mortality, respectively, after adjusting for covariates. Overall, the investigation demonstrated that the association between physical activity and the risk of COVID-19 infection and mortality was dose-dependent with a substantially lower risk of COVID-19 infection in individuals who were engaged in vigorous-intensity physical activity or the highest levels of physical activity.

Thus far, this is the first study to report that the level of physical activity is inversely associated with markedly lower risks of COVID-19 infection and mortality on a nationwide scale, that included the entire South Korean population. This highlights the protective effects of physical activity against the risks of COVID-19 infection and mortality. The findings of this study have critical implications for public health, particularly in the setting of the current global COVID-19 pandemic, with no definitive cure. Thus, it is imperative to encourage the incorporation of physical activity along with social distancing and good hand hygiene to reduce the risk of COVID-19 infection and its related mortality in the general population.

In a recent meta-analysis, higher levels of habitual physical activity lowered the risks of community-acquired infectious disease and mortality [18]. In addition, physically active individuals had a lower risk of infectious diseases, such as community-acquired pneumonia, than their inactive counterparts [19,20,21], and physical activity and regular exercise reduced the risk of upper respiratory infections [7] and influenza-associated mortality [8]. In a Cochran review of a total of 14 trials, although exercise interventions did not reduce the number of acute respiratory infection episodes per person per year, the severity of symptoms measured on the Wisconsin Upper Respiratory Symptoms Survey compared with the controls were reduced [22]. It remains unclear whether physical activity offers protective effects against SARS CoV-2 infection and mortality.

Only two studies, to date, have demonstrated that subjects with lower levels of physical activity and exercise capacity are associated at an increased risk of COVID-19 infection [9,10]. In the former study, physical inactivity, smoking, and obesity were associated with COVID-19-related hospitalisation in a community-based cohort study of 387,109 adults in the United Kingdom [9], suggesting that changes in certain lifestyle behaviours may contribute to a reduction in the health risk associated with COVID-19. Moreover, in the latter study, in a single-centre assessment, exercise capacity as a proxy of physical activity levels was inversely associated with the likelihood of hospitalisation [10]. It should be noted, however, that these previous studies had relatively limited cases of COVID-19-related hospitalisations and did not evaluate the impact of physical activity on the reduced risks of COVID-19 infection and mortality as an outcome. These findings, though consistent with those of the previous studies [9,10], elaborate on the impact of physical activity on COVID-19 infection and mortality by demonstrating an inverse association between physical activity and the risks of COVID-19 infection and mortality using a nationwide, medical Republic of Korea database, including 6228 COVID-19 cases and 125,780 matched controls.

Given the J-shaped curve of the association between intensity of physical activity and the risk of infection [23,24], it remains debatable whether high-intensity physical activity increases the susceptibility to COVID-19 infection. It is well recognised that moderate intensity physical activity reduces the risk of upper respiratory infection [25], whereas high intensity exercises such as marathons increase the risk of upper respiratory tract infection by 2- to 6- fold [26]. Notably, in this study, vigorous intensity physical activity was associated with the lowest risk of COVID-19 infection in a dose-dependent fashion and was associated with a 62% lower risk of COVID-19 mortality. However, the relationship between higher intensity physical activity and the risk of COVID-19 should be investigated further.

In the present study, MVPA, and not light physical activity, was associated with lower risks of COVID-19 infection and mortality, suggesting that physical activity of moderate intensity or higher may confer additional beneficial effects on preventing COVID-19 infections. As previously noted, physical activity reduces cardiovascular mortality in a dose-response fashion [17]. Current physical activity guidelines recommend engaging in physical activity at least equivalent to 500–1000 MET-min/week to reduce the risk of cardiovascular diseases [27]. Based on the linear association between physical activity levels and the risk of COVID-19, recommendations should be adjusted for the prevention of both cardiovascular diseases and COVID-19 infection. More than 1000 MET-min/week of physical activity may thus yield greater protective effects against COVID-19 infection and mortality.

Several potential mechanisms could explain the role of physical activity in reducing the risks of COVID-19 and mortality. Regular aerobic exercise may boost innate immunity and result in enhanced protection against viral infections [25]. In addition, regular physical activity suppresses inflammation and reduces the risk of being overweight and obesity, which are strongly related with the risk of COVID-19 [4]. Moderate exercise in mice decreased the morbidity and mortality associated with herpes simplex virus type 1, and lung macrophages provided the exercise-mediated protection against the virus [28]. Exercise training increases the natural killer cell activity compared with that from pre-training [29]. Exercise also improves the level of pro-inflammatory cytokines and the T helper (Th)1/Th2 cytokine balance, as well as T cell function [30]. Individuals who perform high levels of physical activity are characterised by enhanced vagal tone, which may favourably affect the cholinergic, anti-inflammatory pathway, thereby attenuating the proinflammatory cytokine effects [31]. However, further studies are needed to clarify the relationship between higher levels of physical activity and the risk of COVID-19 infection, as well as how physical activity may decrease the risk of mortality in individuals with COVID-19 infection.

This study investigated the benefits of physical activity according to the baseline characteristics and underlying comorbidities. Overall, the benefit of a lowered risk of COVID-19 from physical activity was affected by central obesity. In a meta-analysis of 75 studies, subjects with obesity were more at a risk of COVID-19-associated morbidity [4]. Metabolic dysfunction, immune impairments, and an increased secretion of inflammatory cytokines from adipose tissue may explain the association between obesity and the risk of COVID-19 [4]. The anti-metabolic and anti-inflammatory effects of physical activity may explain the strong benefits experienced by obese subjects. In this study, central obesity, and not general obesity, significantly increased the beneficial effects of physical activity. This result supports the idea that physical activity should be encouraged for centrally obese subjects.

We acknowledge some methodological limitations to our study. First, although the physical activity assessment used in the national health check-up of this study has previously been validated [16,17], these self-reported survey data are largely dependent on subjective perceptions and memory. Thus, future studies aimed at investigating the relationship between physical activity and the risks of COVID-19 infection and mortality using objective measurement tools, such as an accelerometer, are warranted. Second, individual hygiene (i.e., hand washing) may serve as an important predictor of COVID-19. Despite adjusting for several potential confounders, we did not account for factors such as social distancing, wearing a mask, and hand washing. Since the risk of infection depends largely on contact with the pathogen, engaging in strict precautionary rules during exercise should be kept amid the COVID-19 pandemic. Finally, we recorded a relatively small number of COVID-19 mortalities (*n* = 92). Thus, further studies are needed to investigate the role of physical activity on the severity of COVID-19 infection and mortality in a large population with more severe cases.

In a nationwide population-based case-control study covering the entire South Korean population, regular MVPA was associated with lower risks of COVID-19 infection and mortality, highlighting the importance of maintaining appropriate levels of physical activity (>1000 MET-min/week) along with social distancing and quarantine amid the COVID-19 pandemic.

## Figures and Tables

**Figure 1 jcm-10-01539-f001:**
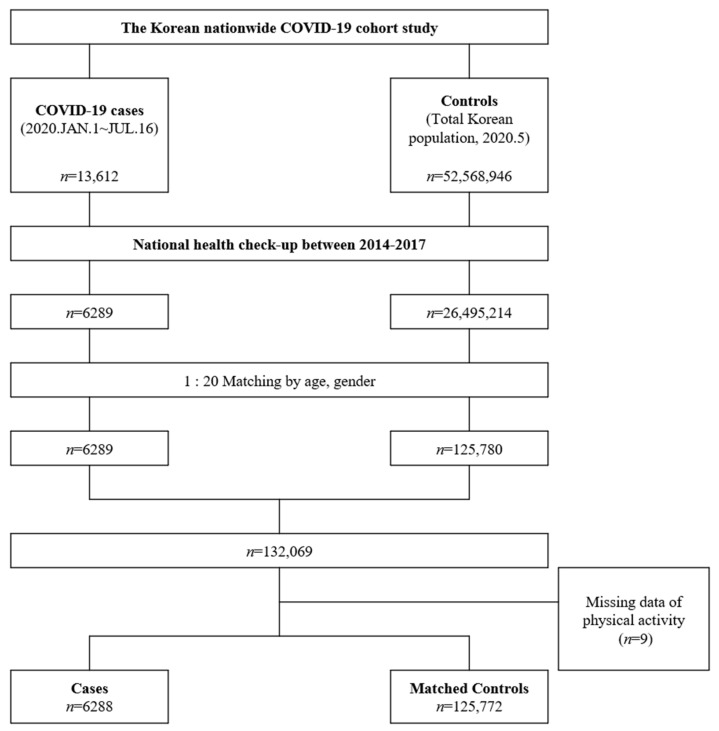
Patients with coronavirus disease (COVID-19) and matched controls. The figure represents a flow diagram of the study.

**Figure 2 jcm-10-01539-f002:**
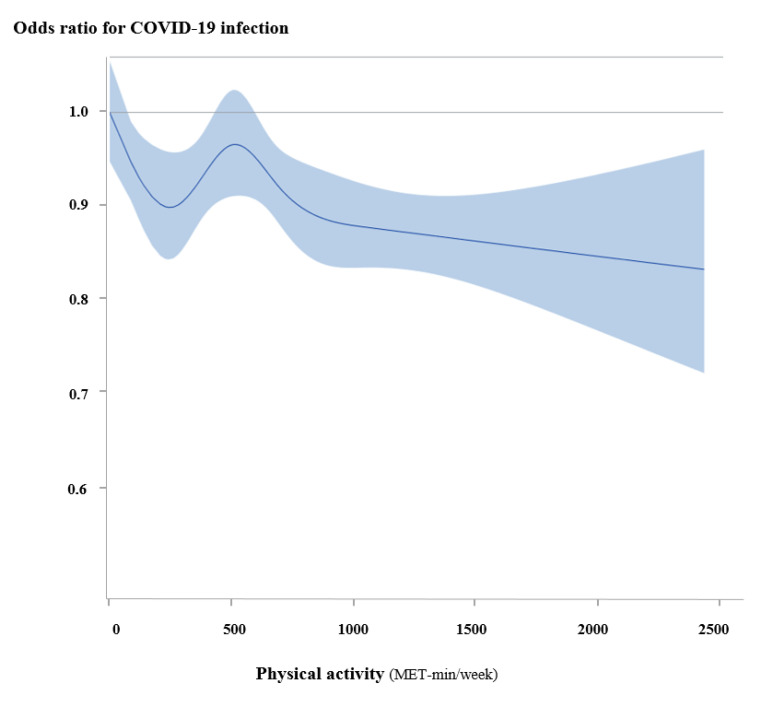
Continuous association between physical activity and COVID-19. According to the level of physical activity, the morbidity of COVID-19 patients linearly decreased.

**Figure 3 jcm-10-01539-f003:**
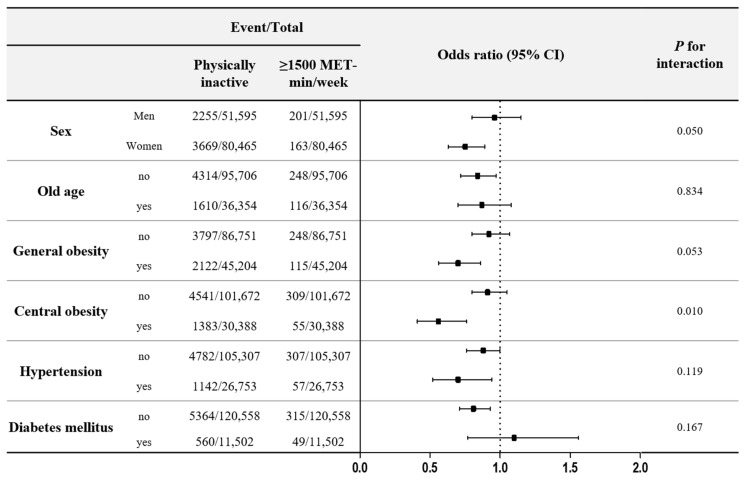
COVID-19 infection and physical activity of the subgroups as stratified by other covariates.

**Table 1 jcm-10-01539-t001:** Demographic and clinical characteristics of case patients and controls.

	COVID-19 (*n* = 6288)	Controls (*n* = 125,772)	*p* Value
Age, years	50.7 ± 14.3	50.7 ± 14.3	0.968
Women, *n* (%)	3832 (60.9%)	76,633 (60.9%)	0.996
BMI, kg/m^2^	24.1 ± 3.4	23.9 ± 3.5	0.003
General obesity, *n* (%)	2237 (35.6%)	42,967 (34.2%)	0.021
WC, cm	80.5 ± 9.6	80.3 ± 14.0	0.079
SBP, mmHg	121.1 ± 15.1	122.0 ± 14.9	<0.001
DBP, mmHg	75.0 ± 10.1	75.6 ± 10.0	<0.001
Medical history, *n* (%)			
Hypertension	1199 (19.1%)	25,554 (20.3%)	0.017
Diabetes mellitus	609 (9.7%)	10,893 (8.7%)	0.005
Dyslipidaemia	571 (9.1%)	10,855 (8.6%)	0.224
Ischaemic heart disease	20 (0.3%)	331 (0.3%)	0.484
Stroke	103 (1.6%)	1692 (1.4%)	0.057
Current smoking	566 (9.0%)	21,507 (17.1%)	<0.001
Alcohol consumption	2511 (39.9%)	62,392 (49.6%)	<0.001
Physical activity, by intensity			
Physically inactive, *n* (%)	1313 (20.9%)	23,978 (19.1%)	<0.001
Light intensity, *n* (%)	1752 (27.9%)	33,185 (26.4%)	
Moderate intensity, *n* (%)	861 (13.7%)	17,557 (14.0%)	
Vigorous intensity, *n* (%)	2362 (37.6%)	51,052 (40.6%)	
Moderate to vigorous intensity, *n* (%)	3223 (51.3%)	68,609 (54.6%)	<0.001
Physical activity, by level of MET-min/week			
MET-min/week	558.2 ± 516.3	580.2 ± 525.7	0.001
<500 MET-min/week	1889 (30.0%)	38,540 (30.6%)	0.001
500–1000 MET-min/week	1973 (31.4%)	38,970 (31.0%)	
1000–1500 MET-min/week	752 (12.0%)	16,340 (13.0%)	
≥1500 MET-min/week	364 (5.8%)	7944 (6.3%)	

Abbreviations: BMI, body mass index; WC, waist circumference; SBP, systolic blood pressure; DBP, diastolic blood pressure.

**Table 2 jcm-10-01539-t002:** Physical activity and the risk of COVID-19 infection (**A**) and mortality (**B**).

**(A)**	**Case** ***n* (%)**	**Control** ***n* (%)**	**Multivariate OR (95% CI)**	
			**Model 1**	**Model 2**
By intensity				
Physically inactive	1313 (20.9%)	23,978 (19.1%)	1	1
Light	1752 (27.9%)	33,185 (26.4%)	0.96 (0.93–1.04)	0.98 (0.91–1.06)
Moderate	861 (13.7%)	17,557 (14.0%)	0.89 (0.82–0.98) *	0.93 (0.85–1.02)
Vigorous	2362 (37.6%)	51,052 (40.6%)	0.84 (0.78–0.90) *	0.88 (0.82–0.94) *
Moderate to vigorous	3223 (51.3%)	68,609 (54.6%)	0.87 (0.83–0.92) *	0.90 (0.86–0.95) *
By level of MET-min/week				
Physically inactive	1313 (20.9%)	23,978 (19.1%)	1	
<500 MET-min/week	1889 (30.0%)	38,540 (30.6%)	0.89 (0.83–0.96) *	0.93 (0.87–1.00)
500–1000 MET-min/week	1973 (31.4%)	38,970 (31.0%)	0.92 (0.86–0.99) *	0.96 (0.89–1.03)
1000–1500 MET-min/week	752 (12.0%)	16,340 (13.0%)	0.84 (0.77–0.92) *	0.86 (0.78–0.94) *
≥1500 MET-min/week	364 (5.8%)	7944 (6.3%)	0.84 (0.74–0.94) *	0.85 (0.75–0.96) *
1 SD of MET-min/week			0.96 (0.93–0.98) *	0.96 (0.93–0.99) *
**(B)**	**Mortality** ***n* (%)**	**Survivor** ***n* (%)**	**Multivariate OR (95% CI)**	
			**Model 1**	**Model 2**
By intensity				
Physically inactive	31 (33.7%)	1313 (21.2%)	1	1
Light	27 (29.3%)	1752 (28.3%)	0.67 (0.38–1.18)	0.57 (0.31–1.04)
Moderate	4 (4.3%)	861 (13.9%)	0.27 (0.09–0.81) *	0.26 (0.08–0.81) *
Vigorous	13 (14.1%)	2362 (38.1%)	0.40 (0.20–0.81) *	0.38 (0.18–0.81) *
Moderate to vigorous	17 (18.5%)	3223 (52.0%)	0.45 (0.25–0.80) *	0.47 (0.26–0.87) *
By level of MET-min/week				
Physically inactive	31 (33.7%)	1313 (21.2%)	1	1
<500 MET-min/week	19 (20.7%)	1886 (30.4%)	0.65 (0.35–1.21)	0.54 (0.28–1.08)
500–1000 MET-min/week	16 (17.4%)	1973 (31.8%)	0.44 (0.23–0.85) *	0.42 (0.21–0.83) *
1000–1500 MET-min/week	6 (6.5%)	752 (12.1%)	0.53 (0.21–1.35)	0.56 (0.21–1.49)
≥1500 MET-min/week	3 (3.3%)	364 (5.9%)	0.30 (0.09–1.03)	0.23 (0.06–0.85) *
1 SD of MET-min/week			0.66 (0.49–0.88) *	0.65 (0.48–0.88) *

Model 1: adjusted for age and sex. Model 2: adjusted for age, sex, income, medical history (obesity, hypertension, diabetes, dyslipidaemia, ischaemic heart disease, and stroke), smoking status, and alcohol consumption. * means *p* < 0.05.

## Data Availability

The current study is not publicly accessible due to Korean personal information projection act. Data are available from the corresponding author on reasonable request.

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
