# Peer review of "Physical Activity and the Risk of COVID-19 Infection and Mortality: A Nationwide Population-Based Case-Control Study"

_jcm, 2021, doi:10.3390/jcm10071539_

Round 1

Reviewer 1 Report

Abstract line 22. “Consecutive” does not quite make sense.

Abstract line 29-30 – correlation not causation, please make this clear.

Introduction:

Line 45: Quarantine is currently the best option.

I caution against the overuse of passive tense, and the overuse of conjunctions at the beginning of sentences.

Methodology:

I believe the choice of statistical methods as appropriate.

Overall:

Good use of flow chart and graphs.

Minor errors in language and spelling, likely missed in the final edit. I would advise an external proofreader who has not seen this before to check through again.

Reviewer 2 Report

Cho et al report on the association between physical activity and the risk of COVID-19 infection and mortality. The manuscript is very interesting, timely, and generally well written, and an important contribution in the current time. They highlight the benefits of physical activity and relevance for population-based prevention. Based on the impressive sample size, results emerge that should be considered under pandemic conditions.

Major Comments:

  1. Methods
    1. Statistical Analysis: It only describes which variables were considered in the multiple regression model. Model 1 and Model 2 are described in the tables. Preferably, describe here that different models were calculated.

  1. Statistical Analysis: Hypertension is described as a continuous variable. In Table 1, hypertension is given as frequency. Therefore, I would characterize hypertension as a dichotomous variable, too.

  1. Results
    1. The decimal places of the p-value are used inconsistently. Mostly 4 decimal places are reported. Sometimes p-values are rounded to 3 decimal places. I would prefer 3 decimal places. Therefore, please revise Table 1. In addition, please unify “p-value for trend = 0.002” in Line 159 and p-value in Table 2. A significant result is reported in line 170 (0.0495). In the context of bringing down to 3 decimal places, this is not a significant result.

  1. Why are some p-values missing in Table 1?

  1. Table 2 - A distinction is made between inactivity and vigorous activity as well as MET levels. What does p-trend refer to? Does it apply to both or only MET-levels? This is not clear and should be revised.

  1. Table 2 – MET-level “≥ 500” and “≥1000” are not described before. Why are they added? Reasons for these categories?

  1. Table 2 – Maybe relevant OR-results could be highlighted

  1. Even if it overlaps, it would be useful to consider the frequency and percentages of cases and controls in Table 2 (A and B).

  1. Please replace chapters 3.3 and 3.4.

  1. Discussion
    1. Line 241-243: “Current physical activity guideline recommends engaging in a physical activity equivalent to 500-1000 MET-min/week in order to reduce the risk of cardiovascular diseases”. These are recommendations for the prevention of cardiovascular diseases. I suppose the authors want to highlight that the recommendations do not apply to the prevention of infection risk and mortality from COVID-19. However, this should be clarified. The recommendations for activity levels (accumulation of 500-1000 MET-min/week) of the prevention of cardiovascular diseases refer to a “minimum amount”. In the context of the present results, recommendations maybe could be adjusted upwards for both, prevention of cardiovascular diseases and Covid-19 infection and mortality.

  1. Overall, I still see a question that needs to be addressed in more depth. The risk of infection depends largely on contact with the pathogen. Therefore, current a review shows “Exercise did not reduce the number of Acute respiratory infection episodes…” (Grande  AJ, Keogh  J, Silva  V, Scott  AM. Exercise versus no exercise for the occurrence, severity, and duration of acute respiratory infections. Cochrane Database of Systematic Reviews 2020, Issue 4. Art. No.: CD010596. DOI: 10.1002/14651858.CD010596.pub3. Accessed 19 March 2021.). This is already reflected by the authors in lines 266-267. However, it should be taken into account a little more.

  1. Line 273 – 276: Recommendations for energy expenditure could be added.

  1. Overall - Little attention is paid to subgroup analysis. Since it does not provide any important findings (all covariates are already included in the multivariate regression model), it could be considered to wave it.

Minor Comments:

  1. Line 31: Abbreviation of SD
  2. Line 115: Capitalize after the dot (Physical…)
  3. Line 116: activities?
  4. Line 120: … sedentary of physical activity = 0 -> maybe “inactivity” instead of activity?
  5. Formatting Table 1 vs. Table 2: Bold fonts are used in Table 2 for variables/ characteristics.
